# An individualized causal framework for learning intercellular communication networks that define microenvironments of individual tumors

**Xueer Chen**[1,2☉], **Lujia Chen**[1,2☉], **Cornelius H. L. Kürten**[3,4,5☉], **Fattaneh Jabbari**[1,2], **Lazar Vujanovic**[3,4], **Ying Ding**[6], **Binfeng Lu**[7], **Kevin Lu**[8], **Aditi Kulkarni**[3], **Tracy Tabib**[5], **Robert Lafyatis**[5], **Gregory F. Cooper**[1,2,4]*, **Robert Ferris**[3,4]*, **Xinghua Lu**[1,2,4]*

1 Department of Biomedical Informatics, University of Pittsburgh, Pittsburgh, Pennsylvania, United States of America, 2 Center for Causal Discovery, University of Pittsburgh, Pennsylvania, Pittsburgh, United States of America, 3 Department of Otolaryngology, University of Pittsburgh, Pennsylvania, Pittsburgh, United States of America, 4 University of Pittsburgh Hillman Cancer Center, University of Pittsburgh, Pittsburgh, Pennsylvania, United States of America, 5 Department of Otorhinolaryngology, Head and Neck Surgery, University Hospital Essen, University Duisburg-Essen, Duisburg, Germany, 6 Department of Biostatistics, University of Pittsburgh, Pennsylvania, Pittsburgh, United States of America, 7 Department of Immunology, University of Pittsburgh, Pennsylvania, Pittsburgh, United States of America, 8 Williamsville North High School, Williamsville, New York, United States of America

☉ These authors contributed equally to this work.
* gfc@pitt.edu (GFC); ferrrl@upmc.edu (RF); xinghua@pitt.edu (XL)

**Data Availability Statement:** We utilized data from public domains. All data are currently publicly

## Abstract

Cells within a tumor microenvironment (TME) dynamically communicate and influence each other's cellular states through an intercellular communication network (ICN). In cancers, intercellular communications underlie immune evasion mechanisms of individual tumors. We developed an individualized causal analysis framework for discovering tumor specific ICNs. Using head and neck squamous cell carcinoma (HNSCC) tumors as a testbed, we first mined single-cell RNA-sequencing data to discover gene expression modules (GEMs) that reflect the states of transcriptomic processes within tumor and stromal single cells. By deconvoluting bulk transcriptomes of HNSCC tumors profiled by The Cancer Genome Atlas (TCGA), we estimated the activation states of these transcriptomic processes in individual tumors. Finally, we applied individualized causal network learning to discover an ICN within each tumor. Our results show that cellular states of cells in TMEs are coordinated through ICNs that enable multi-way communications among epithelial, fibroblast, endothelial, and immune cells. Further analyses of individual ICNs revealed structural patterns that were shared across subsets of tumors, leading to the discovery of 4 different subtypes of networks that underlie disparate TMEs of HNSCC. Patients with distinct TMEs exhibited significantly different clinical outcomes. Our results show that the capability of estimating individual ICNs reveals heterogeneity of ICNs and sheds light on the importance of intercellular communication in impacting disease development and progression.

available. The links to data and code are as follows: The code for iGFCI is available: https://github.com/fattaneh/tetrad-IGFci Single cell RNAseq data are available through GEO database: GSE164690

**Funding:** This work is partially supported by the following NIH grants: R01LM012011 (XL), R01CA254274 (BL), and U54HG008540 (GFC). The funders had no role in study design, data collection and analysis, decision to publish, or preparation of the manuscript.

## Author summary

Intercellular communication network (ICN) within a tumor microenvironment (TME) defines the immune environment of the tumor. Discovering ICNs of individual tumors would shed light on the immune evasion mechanisms of each tumor, enable discovery of potential target of immune therapy, and guide precision immune therapy. An ICN is essentially a causal network, in which changes in cellular state of a cell may causally influence the cellular states of cells surrounding it. In this article, we present a principled Bayesian causal discovery framework for modeling ICNs. The framework involves multiple steps: inferring transcriptomic programs of diverse cells in TME, inferring states of transcriptomic programs of cells in a tumor, learning causal relationships between cellular transcriptomic programs across cell boundary, and finally learning a tumor specific ICN for each tumor. Our study reveals distinct TMEs among HNSCC tumors potentially associated with distinct immune evasion mechanism.

This is a *PLOS Computational Biology* Methods paper.

## Introduction

Communications among cells in a tissue influence their cellular states, leading to a homeostatic state at a given time that impacts the physiological and pathological processes of cells. As an example, cancer cells expressing neoantigens actively modulate the immune environment through intercellular communications [1–6] in order to evade immune surveillance. The successes of contemporary immune checkpoint inhibitors [7–9] demonstrate the potential for identifying and targeting intercellular communications in treating diseases. Importantly, heterogeneity of genomic alterations among tumors also leads to diverse TMEs [10, 11] and thereby to distinct immune evasion mechanisms and heterogeneous responses to immune therapies. Thus, it is important to understand how cells communicate with each other within each tumor and gain insights into tumor specific immune evasion mechanisms to guide personalized immune therapies.

Studying cell-cell communication is a long-standing problem [12–14], and the advent of technologies for profiling single cells [15–20] provides unprecedented opportunities to advance the field. An increasing number of studies concentrate on modeling cell-cell communication through mining single cell transcriptome data [20–26]. However, the majority of current efforts (see recent reviews by Almet et al. [25] and Armingol et al. [26]) concentrate on modeling potential interactions of known ligand-receptors, through studying the correlations between the expression values of ligand-receptor pairs. While informative, a knowledge-based approach cannot discover previously unknown interactions; co-expression of a pair of ligand-receptor mRNA does not entail physical interaction between their protein products; and, more importantly, correlation analysis does not provide information regarding the biological consequence of ligand-receptor interactions. Finally, contemporary modeling of cell-cell communication is mostly performed at a population level, where a single model is developed to explain the data of all cases. Such analysis is not suitable to investigate ICNs underlying the well-appreciated heterogeneity of TMEs observed in tumors [10, 11]. Thus, there is an unmet need for computational methodologies that enable studying ICNs within individual tumors to better understand how cells communicate within a tumor's TME.

An ICN is essentially a causal network, in which a change in the cellular state of a cell (or a subpopulation of cells) can causally influence the cellular states of other cells via communication channels involving physical or paracrine ligand-receptor interactions. If we can accurately detect and represent the state of cellular processes in cells, we can apply principled causal discovery methods to search for causal relationships among the cellular processes across cells, i.e., an ICN. As a discipline of machine learning and artificial intelligence, the field of computational causal discovery has witnessed significant progress in recent decades [27–29]. Particularly, a new family of individualized (or instance-specific) causal discovery methods [30–32] has been developed, which can learn a causal network based on the data of an individual case, be it a single cell or a single tumor.

In this study, we developed and applied an individualized Bayesian causal discovery framework to learn tumor specific ICNs, and we used HNSCC cancer as a testbed for our study. The framework consists of multiple steps as follows. First step, we applied a statistical model to identify GEMs from single cell data, such that expression status of a GEM would reflect the state of a transcriptomic process regulating its expression in a cell or a tumor. Identification of GEMs lays a foundation for modeling how the change of a cellular process in one type of cell (e.g., cancerous epithelial cells) causally influences the state of another process in another type of cells (e.g., CD8+ lymphocytes). Second, we inferred the activation states of the discovered transcriptomic processes in a large number of tumors, so that we could reliably learn causal relationships of cellular processes across different types of cells. Third, we employed causal Bayesian network (CBN) learning algorithms to search for ICNs among diverse types of cells at the cohort and the individual tumor levels. Our analyses uncovered ICNs that model well the coordinated expression of GEMs across diverse types of cells. By mining the patterns of ICNs of individual tumors, we discovered distinct patterns of ICN structure that underlie disparate TMEs of HNSCCs that are associated with significantly different patient outcomes. Our methods serve as a general tool for studying intercellular communications that influence disease development and progression.

## Results

### Identification of context-specific GEMs reflective of the states of transcriptomic processes

We collected single-cell RNA sequencing (scRNAseq) data of 109,879 cells from 18 HNSCC tumors and matched peripheral blood samples. After quality control, we retained data on 78,887 CD45$^+$ cells from tumor and peripheral blood samples of the 18 patients, and we collected 19,107 CD45$^-$ cells from 12 of the tumors (Figs 1A and S1). Since different cells (e.g., epithelial vs lymphocytes) have significantly different transcriptomes and regulatory processes, we separated cells into 4 major categories and modeled their transcriptomic regulatory process respectively: immune-related (PTPRC/CD45$^+$; 78,887 cells), epithelial (9,852 cells), fibroblast (3,476 cells), and endothelial (5,779 cells), based on the expression of respective marker genes.

Based on the assumption that the transcriptome of each single cell is a mixture of GEMs controlled by different transcriptomic processes, we applied the nested hierarchical Dirichlet processes (NHDP) [33] model to the scRNAseq data to identify GEMs. NHDP is a nonparametric statistical model that can deconvolve a mixture of discrete variables (e.g., words in a text documents) by searching for modules (topics) that capture co-occurrence of words in text documents. When applied to scRNAseq data from a population of cells, NHDP can discover sets of co-expressed genes (i.e., GEMs), with each GEM consisting of genes that are repeatedly co-expressed in many cells. As such, the expression status of a GEM in a cell reflects the state of a transcriptomic process that regulates the GEM. NHDP is a hierarchical model

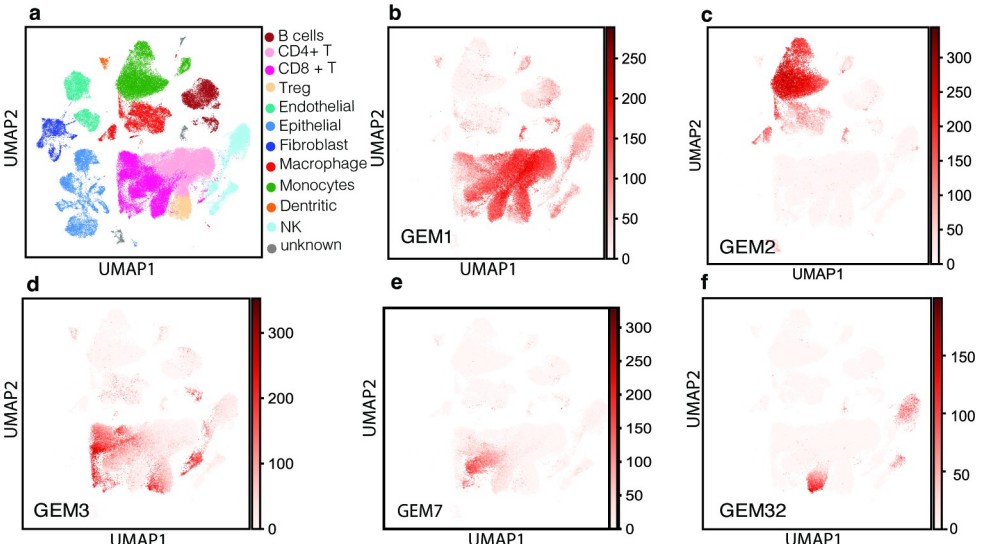

**Fig 1. a. UMAP plot of all cells collected from 17 HNSCC tumors.** Cells were represented in the original transcriptome and were projected to 2-D space using the UMAP method. Pseudo-colors illustrate the cell types. **b–f. Distribution of example immune-cell GEMs among CD45⁺ cells**. The sum of the probabilities of the genes expressed in a cell being a member of an indicated GEM is represented as pseudo-colors.

that represents the joint distribution of GEMs (i.e., co-occurrence patterns of GEMs in cells) using a tree-structured hierarchy. It can represent the differentiation (evolution) processes of cells in a cell population, such that a GEM close to the root of a tree tends to be expressed broadly among the majority of cells, representing a general transcriptomic process shared by cells of a common lineage. In contrast, a GEM close to the leaves of the tree tends to be expressed in a subpopulation of cells, reflecting the state of transcriptomic processes that lead to differentiation of a subtype of cells.

Applying the NHDP model to pooled peripheral and tumor-infiltrating lymphocyte (TIL) CD45⁺ cells, we identified 39 GEMs (referred to as immune GEMs) and their signature genes, and we examined the expression patterns of each immune GEM among the CD45⁺ cells (Figs 1 and S2 and S1 Table). As an example, immune GEM1 is broadly expressed in T cells, B cells, and natural killer (NK) cells, and thus, it represents the transcriptomic process that is shared by cells from the lymphoid lineage; immune GEM2 is mainly expressed in cells of myeloid (monocyte/macrophage) lineage; immune GEM3 is concentrated in TILs, potentially reflecting the impact of tissue environment on T cells [34]; GEM14 is mainly expressed in CD8⁺ T cells, which contain genes commonly expressed in activated cytotoxic T cells (such as *CD8A*, *GZMB*, *CXCL13*, *IFNG*, *GZMA*, *GNLY*, *NKG7*, and *PRF1*), suggesting that this GEM is likely regulated by a T-cell activation processes. Interestingly, certain GEMs can be expressed in cells across conventionally defined immune cell lineages. For example, GEM32 is shared among T cells and NK cells from peripheral blood (Figs 1F and S2), indicating that our approach can discover shared transcriptomic processes among cells that otherwise have distinct characteristics.

To investigate the potential advantage of hierarchical probabilistic topic models in single-cell transcriptomic setting, we compared our results with those from a non-hierarchical probabilistic topic model. We applied the Cellular Latent Dirichlet Allocation (Celda) model [35] to the CD45+ immune cells to identify 30 topics (the same number as our immune GEMs). Celda is a Bayesian hierarchical model, which considers the hierarchical nature of cell

organization in biological systems, i.e., organisms, tissues, cells within a tissue, molecular pathways active in a cell leading to the expression of GEMs, genes co-regulated by a pathway constituting a GEM. However, following the convention of latent Dirichlet allocation [36], at each level of the hierarchy the distribution of components are deemed as independent. For example, the presence (or absence) of GEMs in a cell is deemed to be independent. As such, genes commonly expressed in many cells are distributed to multiple topics (S3 Fig), whereas NHDP model captures them using GEMs close to root (e.g., GEM1 and GEM 2 in Figs 1 and S2). In general, the GEMs identified by Celda exhibit a tendency of overlapping with each other, with fewer GEMs exhibit lineage-specific expression in comparison to those learned by NHDP.

Intercellular communication involves ligand-receptor interactions. We examined whether genes characterizing a GEM (top 50 genes) include ligands or receptors known to be involved in immune responses. Indeed, our results show that many important ligands/receptors are among the top rank genes of GEM (S2 Table), and thus characterize the cells expressing such a GEM. For example, the Immune GEM34 is expressed in a relatively small population of CD8 + cells, which includes well-known ligands and receptors involved in immune responses: *CXCL13*, *TIGIT*, *PDCD1* and *CTLA4* genes. This suggests that the expression of the GEM is likely associated with signals mediated by these proteins.

In addition to PTPRC/CD45[+] cells, we identified GEMs among epithelial, fibroblast, and endothelial cells, respectively, as well. Similarly, some cell-type-specific GEMs represent lineage-defining transcriptomic processes, and some are unique to a small subset of cells reflecting different activation states (S2 Fig). In summary, the NHDP model can identify GEMs that reflect the state of transcriptomic processes in individual cells, enabling us to represent each cell in the space of transcriptomic processes. This representation not only provides a more concise and robust representation of cellular state than is possible with individual genes, but it also serves as an instrument for modeling the causal relationships of transcriptomic processes across cell boundaries.

## Different combinations of active transcriptomic programs define distinct cell subtypes

We set out to examine whether representing cells based on their active transcriptomic processes (expression status of GEMs) can reveal cell subtypes based their cellular states in contrast to representing cells in a gene space. We performed clustering analyses to identify subtypes within each of the following major immune-cell categories: CD8[+], CD4[+], and T regulatory cells (Treg), B cells, NK cells, monocytes and macrophages (S4 Fig). As an example, eight subtypes (clusters) of CD8[+] T cells were identified (Fig 2A), with each exhibiting a

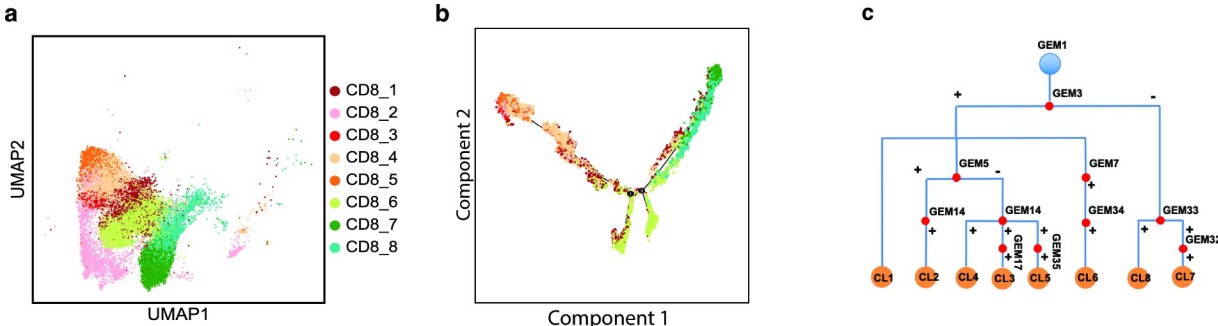

**Fig 2. CD8[+] T cell subtyping based on GEM compositions. a. UMAP plot of CD8[+] T cells.** Eight subtypes were identified, which are indicated by pseudo-colors. **b. Pseudo-time plot reflecting evolution of subtypes of CD8[+] T cells** (cell subtype indicated with the same pseudo color scheme as in **a**). **c. A hierarchical tree reflecting GEM composition of cell subtypes**.

distinct composition of GEMs. Pseudo-time analysis [37] of CD8[+] T cells represented in GEM space revealed a development path of these CD8[+] clusters, starting from "naïve" T cells from peripheral blood, to "primed/activated" tumor infiltrating CD8[+] T cells, and further onto "exhausted" CD8[+] T cells (Fig 2B). Interestingly, the composition of GEMs expressed in these subtypes can be represented as a tree (Fig 2C) that mimics the development path of CD8[+] T cells: All subtypes expressed GEM1, which represents a transcriptomic process shared by cells of the lymphoid lineage; expression of GEM3 differentiates tumor infiltrating and peripheral CD8[+] T cells; expression of other GEMs further leads to differentiation of cells into subtypes. Unlike conventional clustering analysis based on overall transcriptomics, our approach clearly delineated the transcriptomic processes that contributed to such development. For example, cluster CL6 constitutes a subtype of tumor infiltrating CD8[+] T cells that followed a unique trajectory (Fig 2B), and the expression of GEM7 appears to underlie the development of this subtype (Fig 2C). Thus, inspecting the GEM composition of each cell cluster not only reveals distinct patterns of activation states of transcriptomic processes in cells, but also illustrates how a cell subtype develops. Using the same approach, we have identified cell subtypes (clusters) among each major PTPRC/CD45[+] cell category, as well as among epithelial, fibroblast, and endothelial cells (S4 Fig).

## Distinct patterns of active transcriptomic processes and cell composition revealed heterogeneous immune environments of HNSCCs

Based on the assumption that the expression status of a GEM in a tumor would reflect the presence and activation status of transcriptomic process regulating the GEM, we set out to detect the activity of transcriptomic processes by deconvoluting bulk RNA sequencing data of HNSCC tumors profiled by TCGA [38]. Using the top 50 genes of a GEM (ranked according to the probabilities of the genes being assigned to the GEM) as a signature profile of the GEM, we employed gene set variation analysis (GSVA) [39] to determine the expression status of the GEM in each tumor. The GSVA method calculates a sample-wise enrichment score of genes within a GEM and further estimates the variance of GEM enrichment scores across a cohort of samples. Thus, it infers the relative expression status of each GEM, i.e., the relative activation status of the corresponding transcriptomic process in each tumor (S3 Table). We performed GSVA analysis of immune GEMs using the bulk RNA data of 522 HNSCC tumors profiled by TCGA. Similarly, we used the gene expression signature of each subtype of immune cell to infer their relative enrichment scores in the tumors (S4 Table).

We then performed consensus clustering analyses on the tumors using as features the GSVA scores of immune GEMs, and alternatively, the cell subtypes. Both analyses revealed 4 distinct immune environments, with each cluster consisting of distinct compositions of GEMs (Fig 3A) or cell subtypes (Fig 3B), wherein the members of the clusters defined by GEMs and cell subtypes, respectively significantly overlapped. Survival analyses showed significant differences in patient outcomes between the clusters, supporting the impact of the modeled immune environment on disease progression.

Human papillomavirus (HPV) infection causes HNSCC, and it is known that tumors with HPV+ status have distinct immune and clinical phenotypes [40]. We examined distributions of HPV+ patients among the clusters. Interestingly, HPV+ tumors were more enriched in Cluster #4 of GEM based clustering (Chi-square test $p$ = 4.233E-7), suggesting HPV infection may influence the cellular states of immune cells within these tumors. The results also agree with the clinical observations that HPV+ patients tend to have a better survival [38].

To identify the transcriptomic processes impacting survival, we performed Cox proportional hazard analyses, using the GSVA scores of the GEMs (or the cell subtypes) as

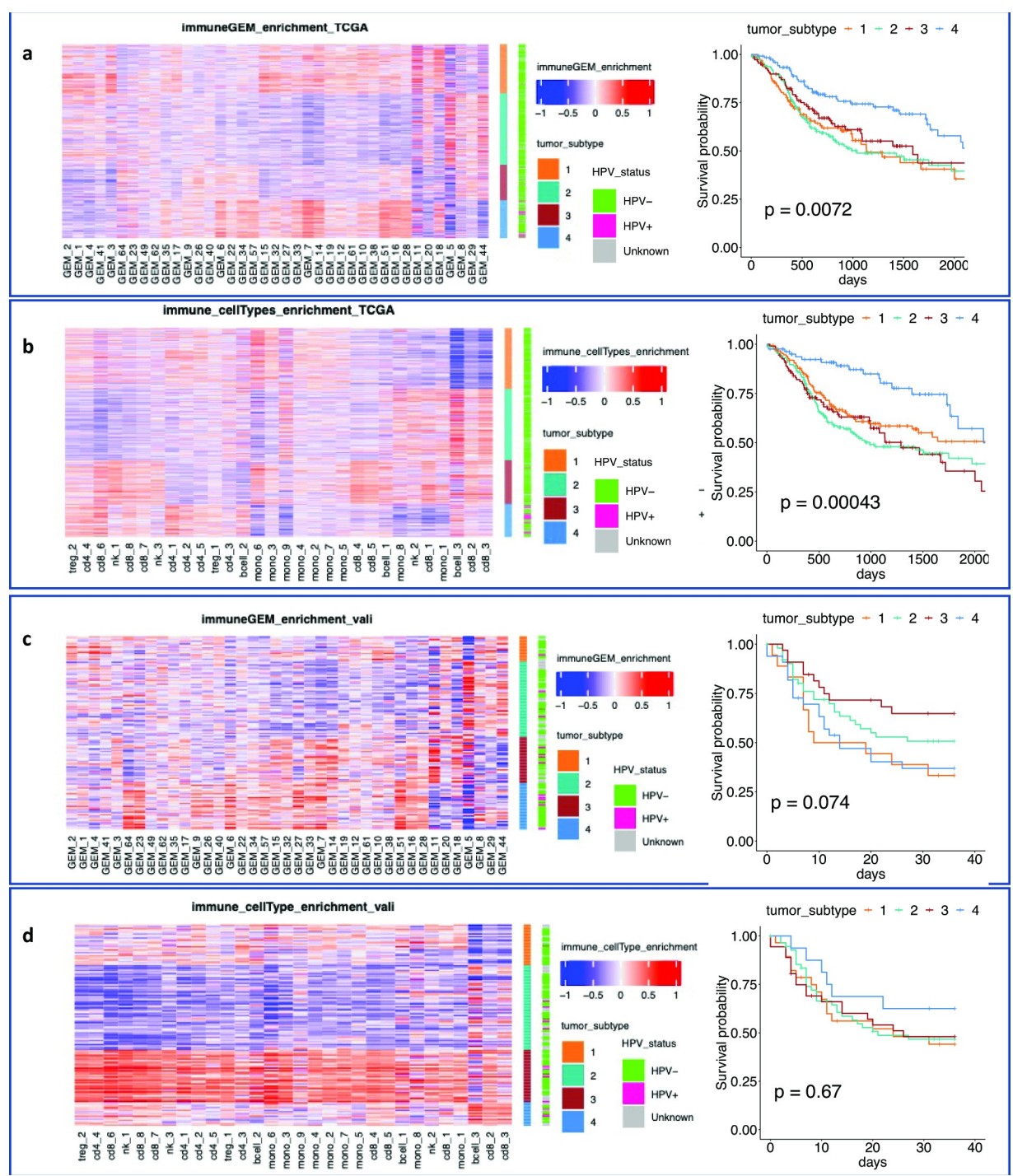

**Fig 3. Distinct immune environment of HNSCCs defined by GEMs and cell subtypes. a.** Heatmap reflecting relative expression levels of GEMs (columns) among TCGA tumors (rows). Tumors are organized according to the clustering results, with color bars that indicate the tumor subtype cluster id. Kaplan-Meier curves of patients belonging to different clusters are shown on the right, using the same color scheme ids. **b.** Heatmap reflecting relative enrichment of immune cell subtypes among TCGA HNSCC tumors. Tumors are organized according to clustering results, and Kaplan-Meier curves of patients belonging to different clusters are shown on the right. **c.** Immune GEMs (with id indicated by the number) that are associated with statistically significant hazard ratios. **d.** Immune cell subtypes (indicated by major cell type, followed by subtype cluster id) that are associated with statistically significant hazard ratios.

independent variables. When only considering the impact of an individual GEM (S5 Table) and immune cell subtype (S6 Table) on survival, 20 GEMs and 17 cell subtypes were deemed to be significantly associated with overall survival of patients ($p < 0.05$). The results of GEM-based and cell-subtype-based analyses are consistent. For example, the expression of immune GEM7 is concentrated in cluster CL6 of CD8$^+$ T cells and the enrichment of the GEM7 or cells from cluster CL6 were deemed as significantly associated with increased hazard ratio; similarly, expression of immune GEM18 is concentrated in cluster CL6 of monocytes (S4 Fig), and their enrichment in tumors were both significantly associated with an increased hazard ratio. The results support that HNSCC tumors exhibit distinct patterns of immune environments, which are associated with patient outcomes.

## Transcriptomic programs of cells are coordinated in TMEs

The capability of detecting the states of transcriptomic processes in cells and individual tumors enabled us to further investigate whether the transcriptomic programs in different cell types are coordinated. We estimated GSVA scores of GEMs of the four major cell types: epithelial, immune (S3 Table), endothelial, and fibroblast cells (S6 Table) and performed pair-wise correlation analyses (Figs 4A and S5). The results showed that expression of GEMs from different types of cells are significantly correlated, supporting the notion that intercellular communication may underlie coordinated activation/inactivation of transcriptomic processes across major cell types. To investigate the impact of cancer and stromal cells on the states of immune cells, we further trained regression models using GSVA scores of the non-immune cell GEMs as independent variables to predict expression of each of the immune GEMs in 10-fold cross-validation experiments. The results show that the expression status (reflected as GSVA scores) of almost all immune GEMs could be accurately predicted (Fig 4B) based on those of non-immune cells. The mean and standard deviation (SD) of $R^2$ for the predicted and observed values of 39 immune GEMs were 0.75 and 0.11, respectively, and $p$-values for all regression $R^2$ (except GEM33) were less than 0.001. The results provide evidence that transcriptomic programs of cells in HNSCC TMEs are coordinated, and such coordination may either be the result of causal intercellular communications, or may be confounded by unknown factors, or both, which can be informed by principled causal analyses.

## GEMs are prognostic of anti-PD-1 responses in HNSCC patients

We further examined whether the GEMs inferred from scRNA-seq would be informative of response to anti-PD-1 immune therapy of tumors, based on the hypothesis that certain GEMs may reflect the states of cells involved in PD-L1/PD-1-mediated immune evasion. We collected the bulk RNAseq dataset of HNSCC patients treated with nivolumab reported by Obradovic et al. [41]. Using the GSVA method, we inferred expression status of GEMs of different cell types, trained elastic net models to predict responses using GEMs as features, and evaluated performance with AUC ROC scores. The patients (n = 48) were randomly split into train (70%) and test (30%) sets. A trained model with GEMs inferred from fibroblast cells (Fib GEM9 and GEM34) showed the best performance (train AUC = 0.81, test AUC = 0.77). These results support the findings by Obradovic et al. that the presence of certain cancer-associated fibroblast cells is prognostic of treatment responses. Using immune GEMs as features, the trained classification model (with Imm GEM5, GEM8, GEM15, and GEM18 as features) achieved good performance as well (train AUC = 0.79, test AUC = 0.70). Whether cells expressing the above GEMs were involved in PD-L1/PD-1-mediated immune evasion remains to be further investigated with a larger sample.

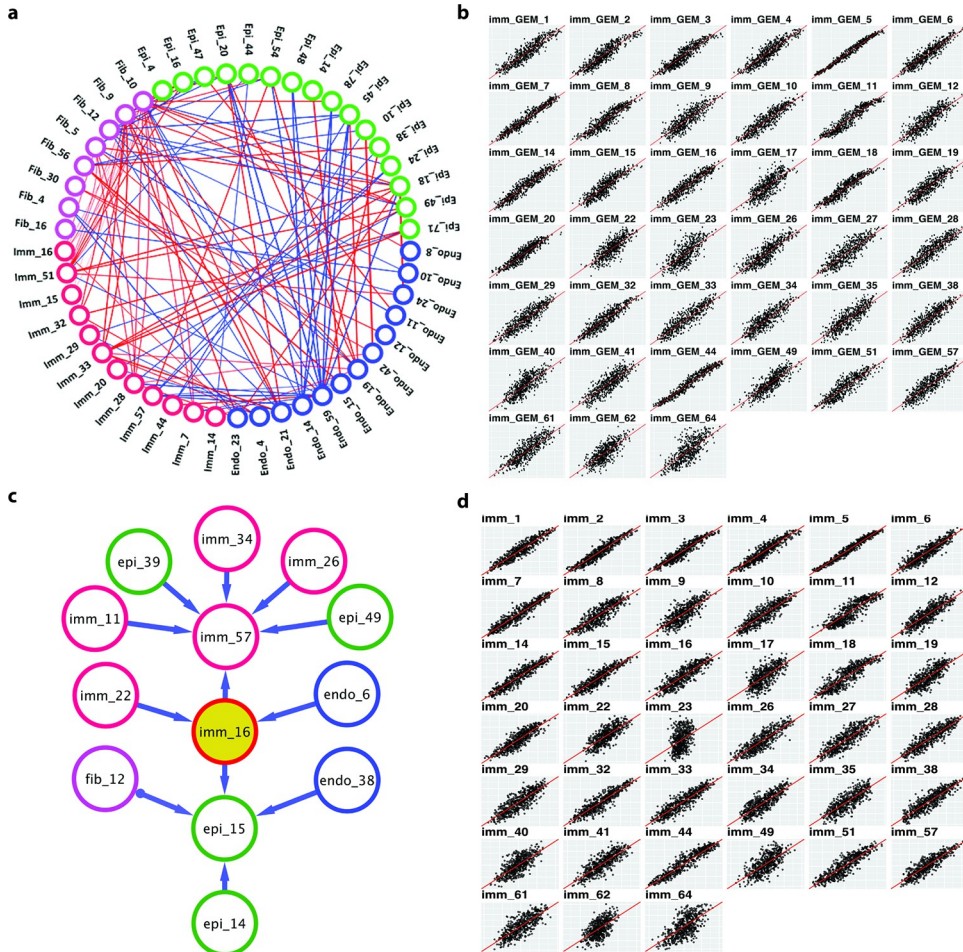

**Fig 4. Intercellular communication among cells in HNSCC TMEs. a. Pair-wise correlation between GEMs from different type of cells.** GEMs from major cell categories are color-coded and the top 10 strongest correlated GEMs for each cell type are shown as edges connecting the GEMs. Positive and negative correlated pairs are indicated as red and blue edges, respectively. **b. Scatter plot of predicted immune GEM enrichment values using non-immune GEMs as independent variables in a regression model vs observed GEMs values.** The average $R^2$ is 0.71. **c. A subgraph of the PAG learned by GFCI algorithm.** GEMs from different cell types are color-coded. A direct edge (A➜B) represents a direct causal relationship, i.e., A causes B. **d. Scatter plot of predicted immune GEM enrichment values using the Markov boundary of a GEM as independent variables in a regression model vs observed GEM values.** The mean of $R^2$ is 0.67, and the standard deviation is 0.16.

## A global ICN of HNSCC identified by causal Bayesian network learning

To go beyond the correlation analysis, we next set out to find direct support for general causal relationships between the transcriptomic processes, i.e., discovering an ICN. We employed the greedy fast causal inference (GFCI) algorithm [42], which is capable of learning causal relationships from observational data (under assumptions), including the possibility of unmeasured (hidden) confounding when two or more variables are caused by some hidden process. The GFCI algorithm takes a set of measured variables (e.g., GSVA scores) as input data and returns a partial ancestor graph (PAG) [43], which represents an equivalence class of causal models (see Methods). In our case, a direct causal edge in a PAG from GEM_A to GEM_B across cell types indicates that the state of the transcriptomic process regulating GEM_A in a subpopulation of cells is estimated to be causally influencing the state of the transcriptomic process regulating GEM_B in another subpopulations of cells.

Using the GSVA scores of GEMs from TCGA HNSCC tumors as inputs to GFCI algorithm, we first learned a consensus PAG (see Methods). To concentrate on intercellular communication, we inspected the expression of GEMs in cell populations based on S2 Fig and identified GEMs that were co-expressed in subpopulations. We then constrained the GFCI algorithm to exclude the candidate edges between GEMs that were known to be co-expressed in same subpopulation of cells, and thus, the returned PAG mainly captured the causal of relationships reflecting intercellular communication. Fig 4C shows a subgraph of the consensus PAG (see Methods), and the complete PAG is shown in S6 Fig and S7 Table. Fig 4C uses the immune GEM16 to illustrate causal relationships with other GEMs, where it is shown at the center surrounded by a set of neighboring GEMs (parents, children, and parents of children, which are commonly referred to as Markov blanket). We examined the directions of the causal edges between major cell categories to provide an overview of how different cells communicate with each other in tumor microenvironment of HNSCCs (Table 1), according to our model. The results indicate dense and multi-way intercellular communications among cells in TMEs.

Since experimentally validating the discovered causal edges is beyond the scope of this study, we set out to mathematically test that the learned ICN precisely captured statistical relationships among the transcriptomic processes in diverse types of cells. Conditional independence plays a central role in discovering causal relationships [27, 28, 44]. In a causal Bayesian network, the value of a variable is independent of the remainder of the variables in the network conditioned on the values of its Markov boundary; in other words, the Markov boundary variables serve as the smallest sufficient set of variables for optimally predicting the state of the GEM of interest. The better we can predict the expression status of a GEM based on the expression status of GEMs in its Markov boundary, the more support (although not proof) it provides for any causal edges within the Markov boundary (as shown in **Fig 4C**) being valid [45]. For each immune GEM, we identified its Markov boundary GEMs (including those from all cell types) and used their expression status (GSVA scores) as independent variables to train a regression model to predict the expression status of the immune GEM of interest. In a 10-fold cross validation experiment, most immune GEMs could be predicted well based on the expression of a few neighboring GEMs discovered by GFCI, with $R^2$ ranging from 0.34 to 0.85 and the mean and standard deviation of the $R^2$ values being 0.63 and 0.13 respectively, and the p-values of regressions (except GEM23) are less than 0.001. Thus, the PAG resulting from our analysis provides a parsimonious representation of an ICN that serves as a hypothesis of the communications between the cells expressing specific GEMs. To test the generalization of the ICN learned from TCGA-HNSC tumor samples, we applied the trained Markov boundary models to an independent HNSC cohort (GSE39366) [35] of 138 tumor samples to predict the immune GEMs. We were able to obtain $R^2$ ranging from 0.36 to 0.74 (p-values: $< 1.4E\text{-}5$), with the mean and standard deviation of the $R^2$ being 0.51 and 0.17.

**Table 1. Statistics of directed causal edges between different cell types.** Rows indicate cell types (endothelial, epithelial, fibroblast, and immune cells) from which edges emit and columns represent cell types to which edges point. A number in the table represents the count of edges between two types of GEMs. (summary of edges in S5 Fig).

| To<br>From | ENDO | EPI | FIB | IMM |
|:---:|:---:|:---:|:---:|:---:|
| ENDO | 25 | 63 | 24 | 38 |
| EPI | 57 | 96 | 35 | 63 |
| FIB | 29 | 34 | 20 | 26 |
| IMM | 28 | 41 | 9 | 90 |

## Tumor specific ICNs of individual tumors underlie the heterogeneous tumor microenvironments of HNSCC

While an ICN trained at the tumor cohort level provides a working hypothesis of the general causal relationships among transcriptomic processes, it does not capture the idiosyncratic characteristics of intercellular communications of individual tumors, which we conjecture underlie the diverse TMEs, as shown in Fig 3. We therefore applied an individualized causal discovery algorithm previously developed by us, which we call the individualized GFCI (iGFCI) algorithm [31] (see Methods) and estimated a tumor specific ICN for each of the 522 HNSCC tumors from TCGA. Based on our current edge selection criteria (see Methods), we noted that GFCI and iGFCI can find a common set of 106 edges, out of 493 edges found by

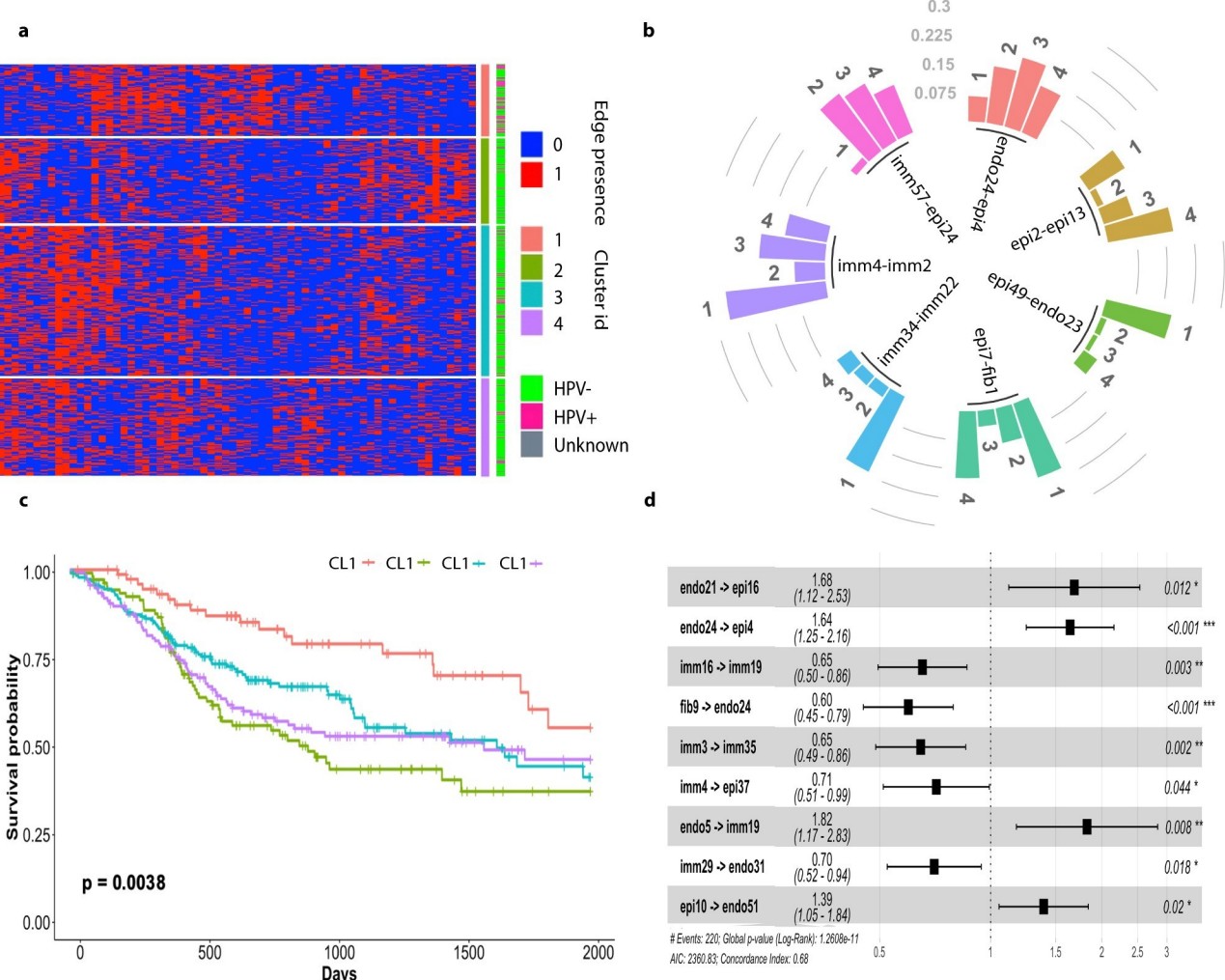

**Fig 5. Distinct intercellular communication networks revealed by individualized CBN discovery. a. Clustering of tumor specific intercellular communication networks.** Tumor specific causal networks returned by the iGFCI algorithm were represented in the space spanned by the union of edges, where the presence/absence of an edge is represented by a binary variable ("1" / "0" respectively). Tumors (rows) were divided into 4 clusters, and the presence/absence of edges (columns) is shown as a heatmap. **b. Examples of causal edges with high variance among tumor clusters.** Eight edges are shown, and the directions of the edges are indexed by the cause-effect GEMs. The height of a bar reflects the ratio (or percentage) of tumors in a cluster (indicated by a number from 1 to 4) that have the edge in their tumor specific network. **c. Kaplan-Meier survival curves of patients assigned to different clusters. d. A list of the causal edges that are associated with a statistically significant proportional hazard ratios according to a Cox proportional hazard analysis.**

GFCI and 549 edges consisting of a union of all ICNs. This indicates that iCFGI may found more individualized causal edges to replace certain population-level statistical relationship.

Inspecting the individual ICNs, we note that while each ICN is unique, certain patterns of network structure were conserved among subpopulations of tumors. We represented each tumor specific network with a vector of causal edges among GEMs, in which element $i$ = 1 if and only if that network contained edge $i$. We then performed consensus clustering analyses of these networks. Four distinct clusters of networks emerged (Fig 5A), with certain edges being distinguishably present or absent from some clusters of tumors (Fig 5B). The patients from the 4 subtypes exhibited significantly different survival outcomes (Fig 5C), even though their clinical characteristics such as tumor stages were similar (S9 Table).

Our results also showed that the presence/absence of the certain edges were significantly associated with patient outcomes (Fig 5D), suggesting that communication between cells may influence the overall immune responses. Of interest, the edge linking immGEM16 and immGEM19, expressed in Treg cells and dendritic cells (DCs) (S7 Fig), respectively, is associated with better prognosis for cancer patients. This result corroborates previous knowledge [46]. Another example is the edge between immGEM3 and immGEM35. ImmGEM3 and immGEM35 were highly expressed in two CD8 T cell subtypes, both of which expressed high-levels of immune checkpoint genes such as PD-1, LAG3 and TIM3 (S7 Fig). This result suggests that communication between these two CD8 T cell subtypes is involved in tumor immune surveillance [47, 48], and such communication is associated with better outcomes.

We further investigated whether the distinct ICN patterns were associated with distinct immune environments, to explore the hypothesis that an immune environment is defined by a subtype type of ICN. We compared the overlap of cluster membership of tumors when they were clustered based on immune GEMs (Fig 3A) and causal network edges (Fig 5A), respectively; we found that the tumor clusters derived from two the clustering analyses share a significant number of tumors (S8 Fig) (chi-square test $p$-value < 2.2E-16). Particularly, it is interesting to note that Network Cluster # 1 closely aligns with the GEM-based Cluster # 4, both were enriched with HPV+ cases, and patients assigned to these clusters have the best outcomes compared to patients assigned to the other clusters. The results suggest the possibility that specific intercellular communication patterns were associated with HPV infection, leading to a distinctive immune environment. In the same vein, Network Cluster #2 aligns best with the GEM-based Cluster #2, and the patients assigned to these clusters exhibited the worst outcomes compared to patients assigned to the other clusters. The results indicate that certain communication channels between different types of cells are highly conserved among tumors, and it is such multi-way communications that may define a TME, which eventually may influence patient outcomes.

## Discussion

In this study, we have developed a framework for estimating tumor specific ICNs, which involves two major conceptual steps: Firstly, mining single cell transcriptomes to identify GEMs to serve as surrogate representations of cellular signaling processes. Secondly, modeling intercellular communication as a causal network between these processes to estimate intercellular communication at the level of individual tumors. We anticipate that the capability of inferring ICN, particularly tumor specific ICNs, will help lay a foundation for future mechanistic investigations of how intercellular communications define the state of cells in a TME, and in turn, how tumors evade the immune surveillance of hosts.

While it is well-appreciated that cells within a TME actively communicate and influence the functional states of each other [19–21], few computational methods have been developed to

analyze the fundamental nature of an ICN, especially the causal effects of communication. Our work represents an effort in this direction. Our results show that the inferred ICNs serve as concise statistical models that explain well the coordinated transcriptomic programs across different types of cells and that tumor specific ICNs provide a new approach to investigate the heterogeneity of immune environments in cancer. Under assumptions (see Methods), the methods we applied identified communication channels (directed edges) of learned ICNs that capture causal effects of intercellular communications, beyond accurately modeling the joint distribution of cellular events (controlled expression of GEMs) in different cells. These assumptions are commonly made in applying graphical causal discovery methods [44]; violations of one or more of them may, however, reduce the accuracy of causal discovery.

However, we realize that those assumptions may not always hold in a complex system as the ones we are studying, and thus, we view the learned ICNs as hypotheses, which can be experimentally tested in the future. A current practical limitation of iGFCI is that it currently operates only on discrete variables. This limitation can be addressed by using more general tests of conditional independence that allow a mixture of continuous and discrete variables.

In this study, we illustrated the capability and utility of our methods in studying ICNs in the cancer (HNSCC) domain. The principled framework introduced here is quite general and can be applied in wide range of biomedical domains to study communications of cells in a tissue or organ. With the advent of more single-cell omics data, particularly in situ spatial omics studies of tissues [15, 49, 50], the framework we present can be used to study intercellular communications under a wide range of physiological (e.g., embryo development) and pathological (e.g., rheumatism or aplastic anemia) settings besides cancer, which provides many opportunities for future research.

## Methods

### Single cell RNA sequencing data processing

FASTQ files were processed with the CellRanger software (10X Inc, version 2.0). We aggregated the 46 samples of all the 18 patients, including samples from CD45+/- tumor specimen and peripheral blood. After the aggregation and default depth normalization using the Cell-Ranger, we collected the count matrix of 103,006 cells in total, with a post-normalization mean reads per cell of 31,303 and median genes per cell of 1,005. We then processed the aggregated count matrix in Scanpy [51]. We filtered cells and genes with cells expressing at least 200 and at most 5000 genes. Cells with more than 5 percent of genes belonging to mitochondria genes were removed. Genes were then filtered with dispersion range (minimum mean of 0.0125, maximum mean of 5 and minimum dispersion of 0.1). Scanpy also provides the implementation of feature reduction and visualization called UMAP [52, 53]. We visualized all the cells using in UMAP in which cells are pseudo-colored to mark cell types based canonical gene signatures of general cell types or manual labels.

### NHDP modeling on single cell RNAseq data

We applied the NHDP on scRNAseq data to identify GEMs, which potentially indicate different biological processes. Unlike non-hierarchical topic models or matrix factorization models [35, 54], the NHDP method allows users to define a hierarchical structure among topics (in our GEMs) and the algorithm will allocate modules along the tree. The NHDP model can capture genes expressed in a majority of cells (aka, house-keeping genes) using topics close to the root, so that the distal topics (GEMs) capture the genes that expressed in distinct subsets of cells. This approach enables the model to mimic the transcriptomic programs underlying cell differentiation processes. In this study, we designed a three-layer hierarchical tree structure,

with branching factors of 5, 4, and 3 from the root to the 2nd and 3rd layers. That is, starting from the root, the 1st layer had 5 nodes; each node in the 1st layer is connected to 4 child nodes in the 2nd layer; each node in 2nd layer is connected to 3 child nodes in the 3rd layer. Each node defines a distribution over the space of genes reflecting the information on which genes are commonly assigned to the module. Input to the NHDP model is a binarized expression matrix of scRNAseq data, where rows are individual cells and columns are genes. An element of the matrix was set to "1" if the corresponding gene is expressed in a given cell, and "0" otherwise. The algorithm generates two outputs: 1) For each node in the hierarchical tree, a distribution over the space of genes is returned. From the distribution, one can extract a ranked list of genes most commonly assigned to the node (GEM). 2) A cell-by-GEM count matrix is returned. An element of the matrix reflects the number of genes expressed in a given cell that are assigned to a specific GEM. This is equivalent to projecting a cell into the space of GEMs, which enabled us to search cell subtypes through clustering analysis. Certain GEMs were not populated by the NHDP, too few genes were assigned to the GEM in cells, and they were excluded from further modeling.

## Enrichment of GEMs in bulk RNAseq and tumor subtype identification

We analyzed the enrichment of all the immune and non-immune GEMs in three bulk RNAseq datasets, namely, TCGA HNSCC tumors and two independent validation sets from the GEO database (GSE39366, GSE27020) [55]. Genes within a GEM are ranked according to the probability mass assigned to them by the NHDP model, and we used the top 50 genes of a GEM to represent its signature co-expression profile. We applied the gene set variation analysis (GSVA) method [39] to estimate the enrichment of a GEM in a tumor relative to a cohort of tumor samples. The distribution of the GSVA scores among the input tumor samples generally followed a normal distribution in the range of -3 to 3.

With the estimated GSVA scores for all the immune GEMs and cell subtypes in TCGA HNSCC tumor samples, we applied consensus clustering (implemented in the R package as ConsensusClusterPlus [56]) and identified subtypes of tumor samples with distinct enrichment patterns of immune GEMs. The optimal number of clusters were chosen using the relative change in area under the empirical cumulative density function (CDF) curve [57], which shows the relative increase in consensus and cluster number at which there is no appreciable increase.

In addition to GEMs, we also calculated the GSVA enrichment scores of immune cell subtypes, which were identified in S4 Fig. The signatures of each cell subtype were curated based on the GEMs that were enriched in the corresponding cell subtypes; in particular, we used the union set of top30 genes of the GEMs enriched in individual cell subtypes. Following the same procedure, we estimated the cell-subtype enrichment GSVA scores in bulk RNA data and performed consensus clustering to identify tumor clusters with distinct composition of cell subtypes.

## Evaluate association between patient survival and GEMs or cell subtypes

We performed Cox Proportional-Hazards regression (*coxph* function of R language) using both the immune GEMs and cell subtypes enrichment scores as input to evaluate which GEMs or cell subtypes were significantly associated with patient survival outcomes. We conducted both individual Coxph regression on individual immune GEMs and alternatively on cell types. To do so, we used a stepwise selection of multiple variates to obtain a minimum and optimized set of immune GEMs or cell types that could fit the survival curves well.

## Cross-cell-type GEM correlations analysis

We examined the correlations among the enrichment scores of GEMs with both Pearson correlation and multiple regression models. Across all cell categories (immune, epithelial, fibroblast, and endothelial), we performed Pearson correlation analysis of the expression levels between all pairs of GEMs. We excluded the intra-cell-category correlation analysis to avoid potential confounding effects of coordinated cell-type-specific expression programs. We also applied the GLMNET [58] package from the R language to train regression models, which predict each immune GEM using non-immune GEMs from epithelial, endothelial, and fibroblast cells as independent variables. We performed 10-fold cross-validation experiments, in which we used 9 folds of data to train a regression model and used the trained model to predict immune GEMs for the tumors in the held-out fold, performing this process over all 10 folds. We then evaluated the performance of the regression models by performing a correlation analysis using predicted GEM-expression values versus observed GEM expression values and recorded $R^2$ values.

## Model intercellular communication network using causal discovery methods

In this study, we set out to model the intercellular communication (i.e., signal transduction across the cell boundary). Here, we introduce the biological problem (Fig 6), which our methods aim to address and how this problem is transformed into a causal Bayesian network problem. We then introduce the computational methods for learning ICNs in detail.

When the state of signaling pathway $X$ changes in a cell (Cell A) (Fig 6A), either due to physiological signal transduction (e.g., signal received from environment, $W$) or pathological perturbation (e.g., a mutation of a gene affecting the pathway, $V$), its signal may be communicated to another type of cell (Cell B) through paracrine or ligand-receptor interaction at the cell surface. The goal of modeling intercellular communication is essentially to model whether the state of the pathway $X$ in Cell A causally influences the state of the pathway $Y$ in Cell B. However, it is usually infeasible to measure the states of pathways (i.e., $X$ and $Y$ are latent variables), making the task of modeling intercellular communication challenging.

To address this challenge, we hypothesize that if changed states of pathway $X$ in Cell A is closely associated with a transcriptomic change of the expression of *GEM-X*, we can use the expression status of *GEM-X* as a surrogate to represent the state of $X$. Similarly, if changed state of pathway $Y$ in Cell B is closely associated with transcriptomic changes of *GEM-Y*, we can use expression status of *GEM-Y* as a proxy to represent the state of pathway $Y$. Under such a setting, we can model the causal relationships between $X$ and $Y$ as a *causal Bayesian network* (CBN) as shown in Fig 6B. Here, $V$, $W$ can be any type of measured variable. For example, they can be the mutation status of a gene, or they can be measured concentrations of signaling molecules. More relevant to this study, they can be extracellular signals transmitted to the Cell A from other cells through cell-cell communication in the same fashion as between $X$ and $Y$. In the latter case, the states of pathways $V$ and $W$ in those cells can be represented by their corresponding GEMs. Under such a setting, we can search for a ICN (i.e., intercellular signal transduction) between latent signaling pathways across cells using GEMs as surrogates of the pathway, like the one shown in S6 Fig.

From a causal modeling perspective, the expression status of *GEM-X* and *GEM-Y* mainly serve as surrogates of the state of pathways ($X$ and $Y$) regulating them. Although it is of value to investigate the functions of genes in each GEM to gain biological insights, interpreting the function of genes in GEMs is not required to model the causal relationships among pathways regulating their expression.

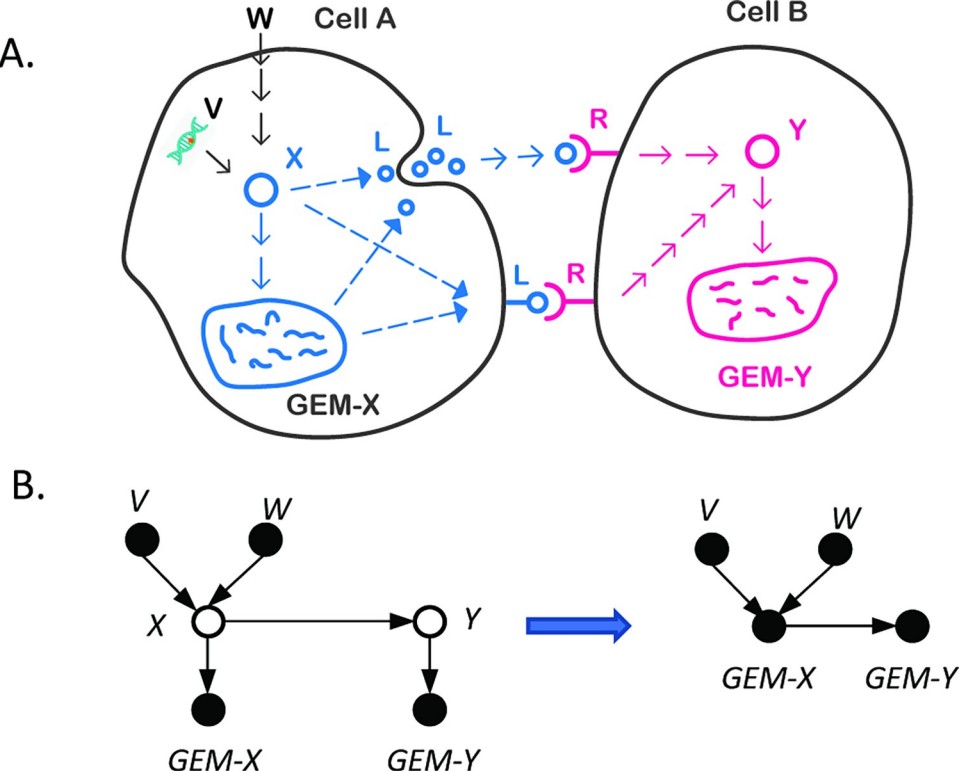

**Fig 6. Causal modeling of an intercellular cellular communication network. A. A biological scenario to be modeled by our framework**. The activation states of the pathway X in Cell A can be affected by physiological signal (e.g., an extracellular signal W) or genomic alteration (e.g., V). Activation of pathway X in Cell A may lead to release or cell-surface presentation of a ligand L, which may be produced through de novo transcription. Ligand-receptor interaction (LR) leads to activation of pathway Y in Cell B. Activation of X and Y may be deterministically associated with expression of corresponding GEMs. **B. A Causal Bayesian network (CBN) for modeling the signal transduction depicted in A**. The CBN on the left side explicitly represents the relationships of pathways X and Y, their upstream signals (V and W), and the GEMs regulated by them. In a CBN, observed variables are represented as filled nodes, and latent variables are represented as open nodes. In general, variables V and W can be any type of observed variable, including the expression values of GEMs reflecting the states of corresponding latent signaling pathways in other cells. Under the above assumption, we can use GEM-X and GEM-Y as proxies of latent variables X and Y, such that the causal edge between X and Y can be represented and estimated as a causal edge between GEM-X and GEM-Y as reported in this work.

## Background on causal modeling and discovery

In our investigation, we used CBNs as the underlying representation of causal relationships. Thus, in this section we first define Bayesian networks (BNs) and then define CBNs as a special version of them. Next, we describe a graphical model called a *maximal ancestral graph* (MAG) that can represent CBNs when they contain unmeasured confounding. It is also possible to graphically model and learn selection bias [59], although we assume no such bias in the data we analyzed in this study, and thus, in this section we do not describe modeling and learning under selection bias. We provide a brief overview of constraint-based causal discovery methods, which test for conditional independence relationships (constraints) in the data and identify a set of MAGs that are consistent with those constraints. We introduce a graphical model called a *partial ancestral graph* (PAG) that graphically represents the set of MAGs that all share the same constraints. If a causal arc appears in a PAG, then all the MAGs consistent with the data-derived constraints contain that causal arc, and thus, that causal relationship is supported by the data.

## Causal modeling methods

A BN consists of a structural model and a set of probabilities [60–66]. The structural model is a directed acyclic graph (DAG) in which nodes represent variables and arcs represent probabilistic dependence. For convenience, we use the terms *node* and *variable* interchangeably in this article. Each node can represent a continuous or discrete variable, although we focus on using discrete variables in this article. For each node $X_i$ there is a probability distribution on that node given the state of its parents, which are the nodes with arcs into $X_i$. The following equation indicates how the node probabilities of a BN specify the joint probability distribution over all $n$ nodes that are being modeled:

$$P(X_1, X_2, \ldots, X_n) = P(X_i | parent(X_i)) \tag{1}$$

The value of a variable may represent any aspect of the modeled entity. The value of any particular variable may be measured or missing. If the value of a variable is always missing, then we say this is an *unmeasured*, *hidden*, or *latent variable*; otherwise, we say it is a *measured or observed variable*. Fig 7 illustrates the structure of a hypothetical BN involving pneumonia and some of its possible consequences.

Since a BN encodes a joint probability distribution, it represents all the information needed to compute any marginal or conditional probability on the nodes in the network model [67].

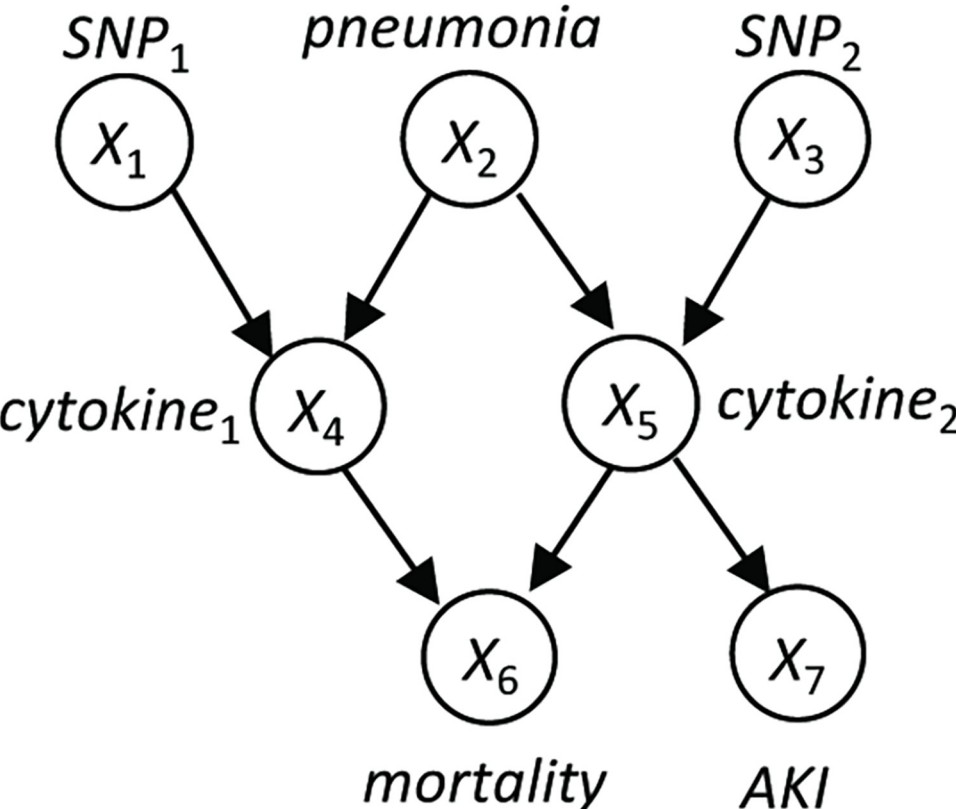

**Fig 7. A hypothetical medical Bayesian network structure with 7 variables represented as nodes.** This structure is presented for illustration only and is not intended to present an accurate and complete model of the causal relationships involving pneumonia and mortality. The variables *cytokine*$_1$ and *cytokine*$_2$ represent two specific types of circulating cytokines associated with inflammatory disease. *SNP*$_1$ and *SNP*$_2$ are particular single nucleotide polymorphisms (SNPs) that represent genetic point variations. *AKI* is acute kidney injury.

A variety of algorithms have been developed for computing $P(A \mid B)$, where $A$ and $B$ are arbitrary sets of variables with assigned values [65, 66, 68, 69].

A *causal Bayesian network* (CBN) is a Bayesian network in which each arc is interpreted as a direct causal influence between a parent node (a cause) and a child node (an effect), relative to the other nodes in the network [64, 70, 71]. A CBN structure encodes the following primary set of independence relationships, which is called the *local causal Markov condition* (LCMC) [71]: Each node is independent of its non-effects (non-descendants), given the state of its direct causes (parents). The local causal Markov condition implies the *global causal Markov condition* (GCMC) which is a graphical set of rules (called d-separation rules) that determine whether $X$ is independent of $Y$ given $\mathbf{Z}$, for every pair of nodes $X$ and $Y$ and every set of nodes $\mathbf{Z}$ [71].

An unmeasured confounder is an unmeasured variable that directly causes two or more measured variables. Unmeasured confounders are common in biological data, and thus, modeling them is important. A MAG [43] is an extension of a CBN structure in which $X \leftrightarrow Y$ represents the presence of an unmeasured confounder between nodes $X$ and $Y$. There is an associated extension of the LCMC that applies to MAGs, as well as an extension of d-separation that is called m-separation [72]. If the node *pneumonia* in Fig 7 was unmeasured, then it would be an unmeasured confounder of *cytokine*$_1$ and *cytokine*$_2$; Fig 8 shows the corresponding MAG.

## Causal discovery methods

We are interested in learning causal relationships from observational data [73], such as the data we analyzed for this article. To do so, we used a causal discovery algorithm that is based

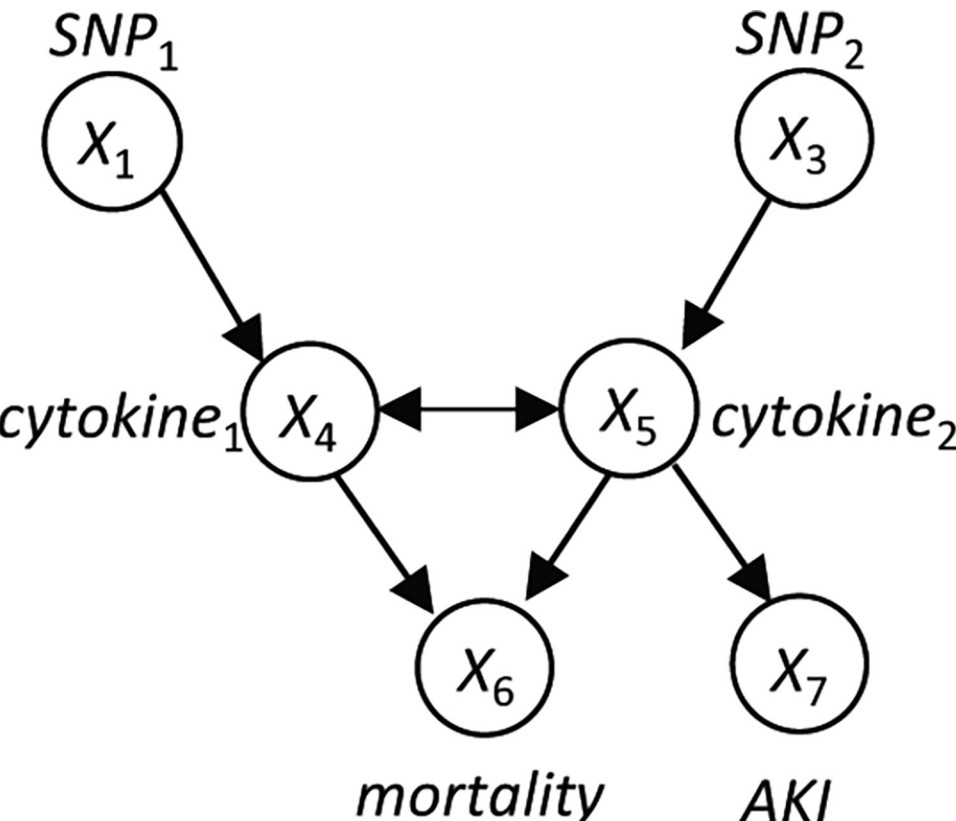

**Fig 8. A MAG representing the causal model in Fig 7 when the node pneumonia is unmeasured.**

**Table 2. A description of the edge types in a PAG and their causal interpretation.**

| Edge type | Relationships that are present | Relationships that are absent |
|---|---|---|
| $X \rightarrow Y$ | $X$ is a cause of $Y$.<br>It may be a direct or indirect cause that may include other measured variables. Also, there may be an unmeasured confounder of $X$ and $Y$. | $Y$ is not a cause of $X$. |
| $X \leftrightarrow Y$ | There is an unmeasured variable (call it $L$) that is a cause of $X$ and $Y$. There may be measured variables along the causal pathway from $L$ to $X$ or from $L$ to $Y$. | $X$ is not a cause of $Y$.<br>$Y$ is not a cause of $X$. |
| $X \circ\rightarrow Y$ | Either $X$ is a cause of $Y$, or there is an unmeasured variable that is a cause of $X$ and $Y$, or both. | $Y$ is not a cause of $X$. |
| $X \circ\!\!-\!\!\circ Y$ | Exactly one of the following holds: (a) $X$ is a cause of $Y$, or (b) $Y$ is a cause of $X$, or (c) there is an unmeasured variable that is a cause of $X$ and $Y$, or (d) both a and c, or (e) both b and c. | |

on the Fast Causal Inference (FCI) algorithm [29, 64]. FCI is one of the most well studied and frequently applied causal discovery algorithms that models unmeasured confounding. Given an observational dataset, it performs a series of statistical tests of the form "*Is X independent of Y given* **Z**", which are called *constraints*. Based on the result of one test (and of other previous tests), FCI will ask for the result of another test. The algorithm has a highly efficient method for performing a series of statistical tests, such that the total set is relatively small. It also starts with low dimensional sets **Z** in order to maximize statistical testing power. FCI continues to perform such tests until no additional testing will further resolve the causal relationships among the measured variables. It then returns a set **C** of causal models (MAGs) that all satisfy the constraints that were tested. Each of the causal edges $X \rightarrow Y$ that appear in all of the MAGs in **C** are supported by the data as being present. FCI outputs a compact, graphical representation of the edges of the MAGs in **C**, which is called a PAG [43]. There are two primary edge types in PAGs: $X \rightarrow Y$ ($X \leftarrow Y$ is possible as well, but does not illustrate a fundamentally different edge type) represents causation, and $X \leftrightarrow Y$ represents unmeasured confounding. There are also two additional edge types in PAGs: $X \circ\rightarrow Y$ represents that the relationship could be either $X \rightarrow Y$ or $X \leftrightarrow Y$, and $X \circ\!\!-\!\!\circ Y$ represents that the relationship could be either $X \rightarrow Y$, $X \leftarrow Y$, or $X \leftrightarrow Y$. The absence of an edge between $X$ and $Y$ denotes that none of these relationships are modeled as existing. Table 2 provides a more detailed description of the interpretation of edges in a PAG; this table originally appeared in an appendix written by Peter Spirtes in the article available at [74].

As an example, consider that the causal process generating the observational data available to us is accurately modeled by a CBN with the structure shown in Fig 7, where pneumonia is an unmeasured variable. Suppose that an analysis of the data yields constraints consistent with that structure. The MAGs consistent with those constraints are shown in Fig 9. The PAG representing those four MAGs is shown in Fig 10. That PAG contains all of the causal relationships among the measured variables in Fig 7, except that the relationship between each SNP and its associated cytokine includes several possibilities; nonetheless, the PAG represents that each SNP <u>may</u> cause its associated cytokine and the cytokine does not cause the SNP.

Even in the best of circumstances (see Causal Discovery Assumptions below), it often is not possible to learn from constraints many of the causal relationships among a set of measured variables. Nonetheless, identifying just a few causal relationships that are novel and valid may be helpful in supporting scientific inquiry. S6 Fig shows that the causal discovery method we applied output many possible causal relationships that serve as causal hypotheses. In general, the FCI algorithm, and many of its extensions, such as GFCI (see below), have been proven under assumptions to discover only and all the causal relationships that can be discovered

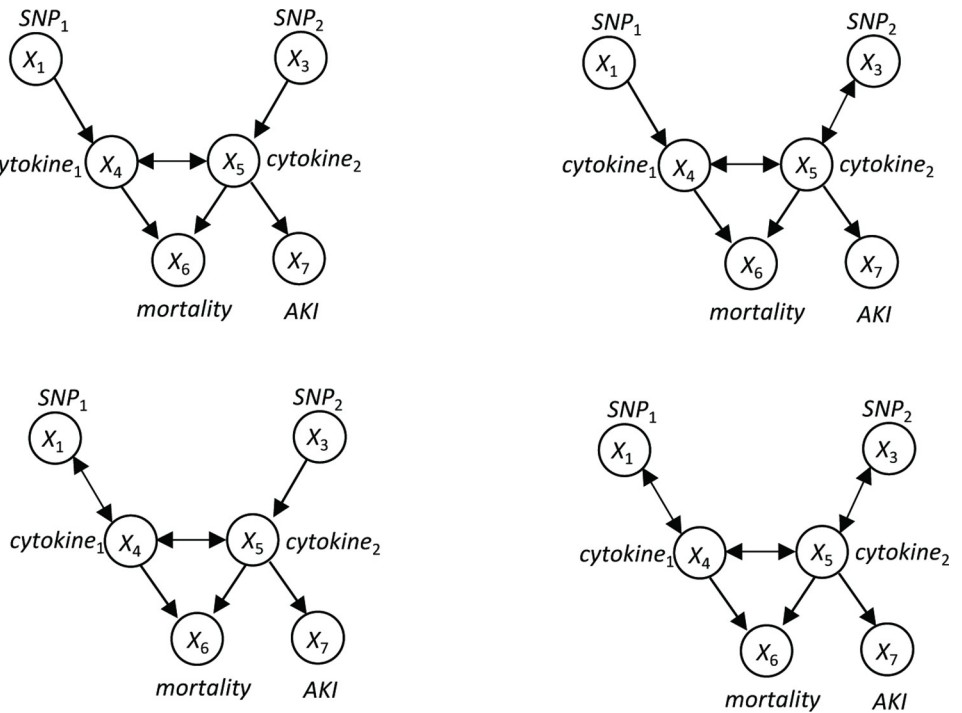

**Fig 9. The MAGs consistent with the CBN shown in Fig 7, assuming *pneumonia* is an unmeasured variable.**

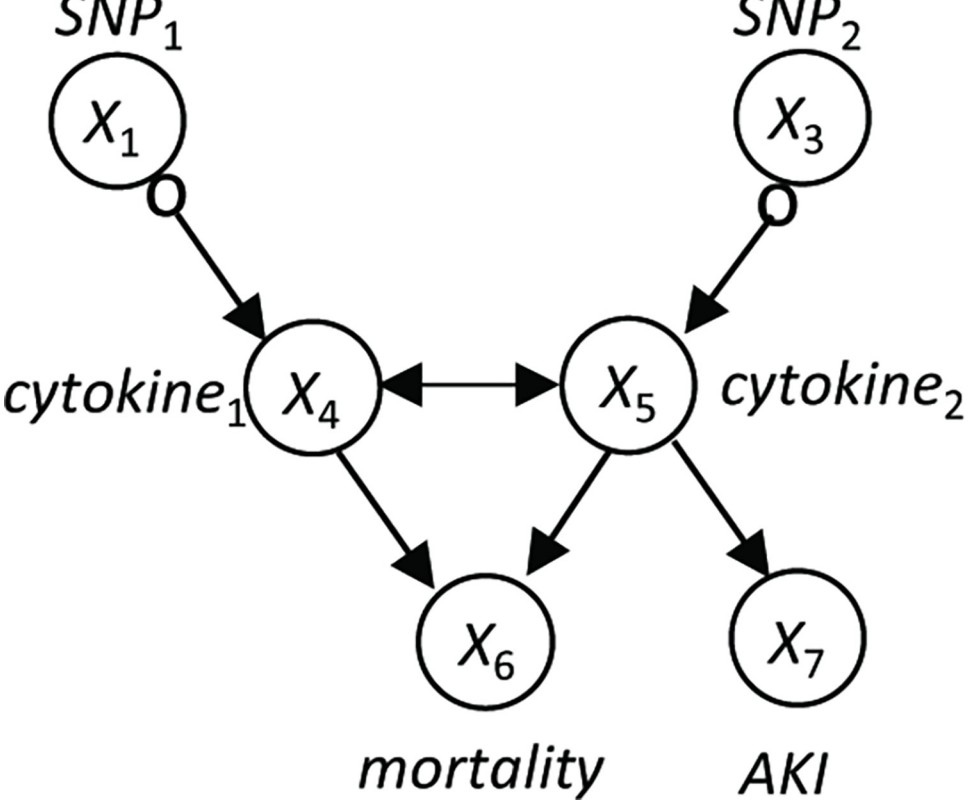

**Fig 10. A PAG representing the four MAGs shown in Fig 9.**

from valid conditional independence constraints [43], and in that sense they are sound and complete.

Among causal discovery methods in use in biomedicine, the approach used by FCI is very similar to the one used by the more recently developed Causal Inference Test (CIT) method [75, 76]. CIT is a constraint-based method that typically evaluates four constraints on three variables: a genomic variable (e.g., a SNP), a variable that represents an intermediate cellular state or process (e.g., an expression variable), and a variable that represents a phenotype of interest. A search of Google Scholar on 01/16/2022 using the phrase "causal inference test" yielded over 300 articles, including analyses of observational biomedical data for causal relationships in obesity, rheumatoid arthritis, psoriasis, ovarian cancer, autism, type 2 diabetes, schizophrenia, Alzheimer's disease, and other diseases and conditions. FCI also can be applied using only three variables, in which case it operates very similar to CIT. More generally, however, FCI can handle causal discovery using thousands of variables. In doing so, it identifies potentially complex patterns of constraints among those variables that support causal relationships that cannot be found using only three variables. In seeking to discover new causal relationships, researchers have applied FCI and closely related algorithms to clinical data in disease areas that include for example alcoholism [77], Alzheimer's disease [78], arthritis[79], and schizophrenia [80]

In the study reported here, we applied the GFCI algorithm [81], which is an extension of FCI. The first phase of GFCI learns a BN structure (DAG) (more precisely, it returns a Markov equivalence class of BNs that cannot be distinguished by independence constraints) that does not model unmeasured variables, using a run-time-optimized version [82] of the GES algorithm [83]. That BN is used to constrain the relationships among the measured variables considered by FCI. Although GES is a greedy algorithm, under reasonable assumptions it has been proven to return the correct BN structure if given sufficient data [83]. Although no statistical algorithm can be guaranteed to return a correct model when given finite data, as is always the case in reality, having an algorithm that is provably correct in the large sample limit is generally considered to be an important property for it to have. In the second phase of GFCI, it applies the FCI algorithm, which begins its search relative to the constraints found by GES in the first phase.

For any method to discover causal relationships from observational data, assumptions are required. For GFCI, the key assumptions are as follows: (1) The causal process being modeled can be represented by a CBN, which possibly contains unmeasured variables. Thus, the structure of the causal relationships can be represented by a DAG. (2) The available observational data can be viewed as having been generated from sampling the joint probability distribution of some CBN. Thus, there is no selection bias due to some types of cases being selectively missing. (3) The independence constraints represented by a CBN structure imply independence constraints in the probability distribution represented by the CBN. 4) The dependence constraints represented by a CBN structure (i.e., where independence constraints do not occur) imply dependence constraints in the probability distribution represented by the CBN. (5) Statistical tests of independence on the available observational data are correct, relative to the independence constraints represented by the data-generating CBN.

Statistical testing on finite data is imperfect, and thus, so are the constraints provided to GFCI. Therefore, as a form of robustness analysis, we applied GFCI multiple times using bootstrapping [84]. For each of $m$ bootstrap datasets of the observational data, we ran GFCI to generate a PAG. Over the $m$ PAGs produced, for each pair of measured variables we report a causal relationship for that pair if that relationship occurred in at least 20% of the bootstrap samples.

## Application of causal discovery methods

We employed the GFCI algorithm to infer from data whether one GEM is supported as causally influencing the expression of another GEM, particularly across cell types. We discretized GSVA scores of a GEM to 0, 1, 2, based on the following percentile brackets: [0–25%], (25% - 75%], and (75% - 100%]. For estimating a population-wide ICN, we applied the discrete version of GFCI algorithm implemented in the Tetrad Package [85]. Unmeasured confounders are likely to be common at the cellular level, and thus, the ability of GFCI to model them is important. On the other hand, there may be patterns of association among GEMs that strongly support that one GEM causes the expression of another; we are particularly interested in analyzing the data for such results.

The Tetrad implementation of the GFCI algorithm allows a user to input prior knowledge regarding the network, such as edges to be included, as well as those to be excluded. To avoid confounding effects of cell-type-specific co-expression of GEMs, we first examined the distribution of GEMs in cells based on the results shown in S2 Fig to identify a list of pairs of GEMs that were co-expressed in a set of cells. This list was input to GFCI as prior knowledge, so that the algorithm would exclude edges between a pair of GEMs in the list. Using the GEM data of TCGA HNSCC tumors as inputs, we produced 50 bootstrap samples by drawing 90% of the tumors each time with replacement. The first phase of GFCI used the Bayesian information criterion with default settings and a penalty discount of 1.0 (see "sem-bic" in the Tetrad manual [86]), and the second phase used the Fisher-$z$-test with alpha = 0.01. We created a consensus causal PAG by keeping the edges that were conserved more than 20% of the time, according to the bootstrap results. There were 15 cases in which both X➜ Y and X ← Y occurred more than 20% of the time in the bootstrap results. In those cases, we did not enter these relationships into the consensus PAG.

## Discovering tumor specific intercellular communication networks

This section first provides background on the methods we developed for performing individualized causal modeling and discovery. It then describes how we applied those methods to the study data to learn tumor specific ICNs.

## Background on individualized causal modeling and discovery

In our investigation, we also used a method that learns causal models that are individualized or specialized to a given tumor. In doing so, we seek to learn more accurate models of the causal relationships in individual tumors and how these relationships vary among tumors. The method we developed [87] and applied considers the values **V** of the variables that characterize a current tumor for which we wish to learn a causal model. In performing constraint-based causal discovery, the method uses those values and a training set of data on other tumors to learn constraints that are individualized to **V**. Those individualized constraints then lead GFCI to learn an individualized causal model; this individualized version of GFCI we call iGFCI. In this section, we provide a high level overview of the iGFCI algorithm; additional details are available in [87].

iGFCI uses variations of the GES and FCI algorithms used in GFCI. As discussed above, GES learns a BN. The individualized version of GES [88], which is used in iGFCI, learns a BN that represents context-specific independence (CSI) [89] wherein the variables that are the parents of a node depend on the values of **V**. Fig 11 shows an example of CSI in a BN. Suppose that $T = \{X_1 = 1, X_2 = 1, X_3 = 0\}$. For this instance $T$, Fig 11B indicates that $X_4$ is independent of $X_3$ given $X_1 = 1$ and $X_2 = 1$; this independence implies that for this instance, the parents of $X_4$ are just $X_1$ and $X_2$, as shown in Fig 11B. Fig 11C shows the PAG that can be learned in the

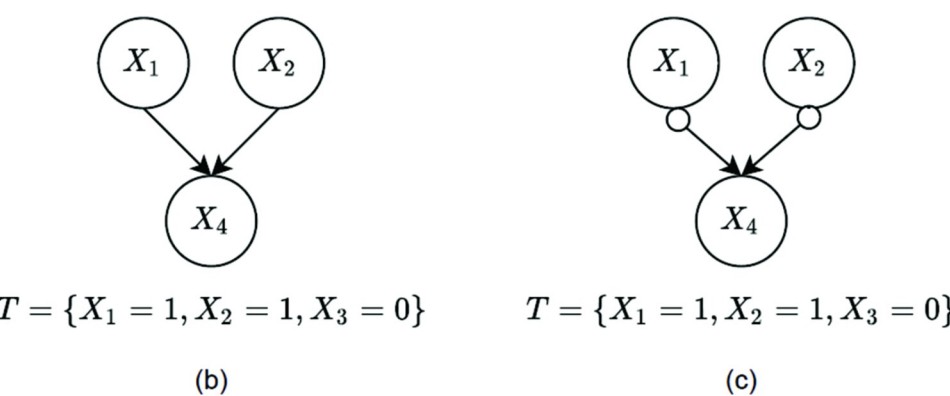

| $X1, X_2, X_3$ | $P(X_4|X_1, X_2, X_3)$ |
|---|---|
| $0, 0, 0$ | 0.90 |
| $0, 0, 1$ | 0.90 |
| $0, 1, 0$ | 0.90 |
| $0, 1, 1$ | 0.90 |
| $1, 0, 0$ | 0.32 |
| $1, 0, 1$ | 0.78 |
| $1, 1, 0$ | 0.13 |
| $1, 1, 1$ | 0.13 |

(a)

$T = \{X_1 = 1, X_2 = 1, X_3 = 0\}$

(b)

$T = \{X_1 = 1, X_2 = 1, X_3 = 0\}$

(c)

**Fig 11. A simple, hypothetical example of CSI.** (**a**) The data-generating CBN for variable $X_4$ given its parents $X_1$, $X_2$, $X_3$; (**b**) the individualized CBN that represents the CSI relationship $X_4 \perp X_3 | (X_1 = 1, X_2 = 1)$; (**c**) the individualized PAG that is learnable in the large sample limit from observational data generated by the CBN in **b**, given the assumptions described above.

large sample limit from data generated by the CBN shown in Fig 11B, under the assumptions discussed above; with additional measured variables or background knowledge, it may be possible to further resolve the causal relationships among these three variables.

The second phase of iGFCI applies a modified version of FCI. The traditional version of FCI tests whether $X$ is independent of $Y$ for all values of **Z**. In contrast, iGFCI includes a context-specific test of conditional independence that assesses whether variable $X$ is independent of variable $Y$ given **Z** = **z**, where **z** are the values given by **V** for the variables in conditioning set **Z**. In this way, iGFCI learns a PAG that is individualized with respect to the values of the variables that characterize tumor T.

The closest prior work to iGFCI is a method reported in Cai et al [30] to learn tumor specific genomic drivers from data. However, that method is limited to searching over bipartite causal graphs on binary variables in which one partition contains causes and the other contains effects. Also, the method assumes there is one and only one cause for each effect. Both assumptions are reasonable for that application, but restrict generality. The iGFCI method is able to learn unrestricted, instance-specific CBNs.

## Application of an individualized causal discovery method

To investigate the mechanisms underlying heterogeneous immune environments among tumors, we applied iGFCI to the GEM variables derived from bulk TCGA HNSCC tumors, as described above. Since the GEMs are continuous variables and iGFCI currently operates on discrete variables, we discretized the GSVA scores of a GEM to 0, 1, 2, based on the following percentile binnings: [0–25%], (25% - 75%], (75% - 100%].

We used bootstrap sampling to enhance the robustness of the results generated by iGFCI. In particular, for each tumor, we applied iGFCI to 20 bootstrap datasets that were generated from the original dataset, where each bootstrap dataset was obtained by 90% sampling of the original dataset with replacement. Using the bootstrap results for a given tumor $T$, we derived the conserved edges of type $X \rightarrow Y$. In particular, we kept the edges that were present in more than two bootstrap-derived PAGs from among the 20 individualized PAGs derived for tumor $T$. The union of the retained causal edges yielded an individualized, "consensus" intercellular communication network (ICN) for each tumor $T$. Specifically, an element of the causal-edge ICN vector for a tumor T was set to "1" if the causal edge was present in the tumor specific intercellular communication network for $T$, and set to "0" otherwise.

## Finding patterns among tumor specific ICNs

We clustered the tumor samples based on their individual ICNs learned using iGFCI to obtain novel subtypes of HNSCC with different cellular communication patterns. Each tumor is represented as a vector over the space of the union of causal edges of the tumor specific intercellular communication networks as described in the previous subsection. We applied the ConsensusClusterPlus package from the R Bioconductor Consortium to cluster these vectors and grouped tumors based on their cluster assignments.

## Supporting information

**S1 Fig. Distributions of cells with respect to cell type and sources. a.** UMAP visualization of major types of cells. **b**. Visualization of cells from different tumors in UMAP 2-D space. C. Distribution of cells from tumors and peripheral blood. D. Distribution of cells from tumors with different human papillomavirus status.
(TIF)

**S2 Fig. UMAP distribution and top 20 genes of GMEs.**
(PDF)

**S3 Fig. UMAP of gene modules learned from CD45+ immune cell using Celda.**
(TIF)

**S4 Fig. Cell subtypes within major cell categories.** Cells were first divided into 9 major categories based on the expression of markers. Within each major category, cells were represented in the GEM space and analyzed using consensus clustering. Each block shows the subtyping within one of 9 major categories.
(TIF)

**S5 Fig. Heatmap representing significant correlation coefficients of pair-wise correlation analyses between non-immune GEMs and immune GEMs.** Value of correlation coefficients are color-coded.
(TIF)

**S6 Fig. The cohort-wise PAG learned by GFCI algorithm.**
(TIF)

**S7 Fig. Cell types expressing different GEMs. A.** UMAP projections of Treg markers (*CD3E*, *CD4*, *FOXP3*) and immGEM16. **B.** UMAP projections of dendric cell markers (*XCR1*, *CLEC9A*, *CD1C*) and immGEM19. **C & D**. UMAP projection of CD8 cell markers (CD3E, CD4, CD8A), immGEM3 and immGEM35 respectively. **E**. UMAP projections comparing distributions of immGEM3 and immGEM35 among CD8+ cells. **F**. CD8+ cell clusters. **G.** Dot plot show the enrichment of immGEM3 and immGEM35 among the subtypes of CD8+ cells.
(TIF)

**S8 Fig. Heat map representing agreements of tumor cluster assignment by comparing clustering results based on GEM expression (columns) and on intercellular communication edges (rows).**
(TIF)

**S1 Table. The top 50 genes of GEMs from immune, epithelial, endothelial, and fibroblast cells.**
(XLSX)

**S2 Table. Lists of known ligand and receptor genes that are among top 50 genes of GEMs from diverse cell types.**
(XLSX)

**S3 Table. Inferred expression values (GSVA scores) of immune GEMs of the TCGA HNSCC tumors.**
(CSV)

**S4 Table. Inferred enrichment values (GSVA scores) of immune cell subtypes of the TCGA HNSCC tumors.**
(CSV)

**S5 Table. A list of immune GEMs with statistical significances by Cox proportion.**
(CSV)

**S6 Table. A list of immune cell subtypes with statistical significances by Cox proportion.**
(CSV)

**S7 Table. Inferred expression values (GSVA scores) of GEMs for epithelial, endothelia, and fibroblast cells.**
(CSV)

**S8 Table. A list of causal edges inferred by GFCI algorithm shown in S6 Fig.**
(CSV)

**S9 Table. Clinical phenotypes of 4 types of subtypes derived based on ICNs.**
(DOCX)

## Acknowledgments

We thank Peter Spirtes and Bryan Andrews for comments on the text.

## Author Contributions

**Conceptualization:** Xinghua Lu.

**Data curation:** Xueer Chen, Lujia Chen, Cornelius H. L. Kürten, Lazar Vujanovic, Binfeng Lu, Kevin Lu, Aditi Kulkarni.

**Formal analysis:** Xueer Chen, Lujia Chen, Cornelius H. L. Kürten.

**Funding acquisition:** Robert Ferris, Xinghua Lu.

**Investigation:** Robert Ferris, Xinghua Lu.

**Methodology:** Xueer Chen, Lujia Chen, Fattaneh Jabbari, Ying Ding, Gregory F. Cooper, Xinghua Lu.

**Project administration:** Xinghua Lu.

**Resources:** Cornelius H. L. Kürten, Lazar Vujanovic, Aditi Kulkarni, Tracy Tabib, Robert Lafyatis, Robert Ferris.

**Software:** Xueer Chen, Lujia Chen, Fattaneh Jabbari.

**Supervision:** Gregory F. Cooper, Robert Ferris, Xinghua Lu.

**Validation:** Binfeng Lu, Kevin Lu.

**Visualization:** Xueer Chen, Lujia Chen.

**Writing – original draft:** Xueer Chen, Gregory F. Cooper, Xinghua Lu.

**Writing – review & editing:** Lazar Vujanovic, Gregory F. Cooper, Xinghua Lu.

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
