## [Decision Letter · Decision Letter 0]

27 Jul 2022

Dear Dr Lu,

Thank you very much for submitting your manuscript "An individualized causal framework for learning intercellular communication networks that define microenvironments of individual tumors" for consideration at PLOS Computational Biology.

As with all papers reviewed by the journal, your manuscript was reviewed by members of the editorial board and by several independent reviewers. In light of the reviews (below this email), we would like to invite the resubmission of a significantly-revised version that takes into account the reviewers' comments.

We cannot make any decision about publication until we have seen the revised manuscript and your response to the reviewers' comments. Your revised manuscript is also likely to be sent to reviewers for further evaluation.

Sincerely,

Philipp M Altrock, Ph.D.

Guest Editor

PLOS Computational Biology

Douglas Lauffenburger

Deputy Editor

PLOS Computational Biology

Reviewer's Responses to Questions

**Comments to the Authors:**

Reviewer #1: The authors discuss how cellular states of cells in TMEs are coordinated through intercellular communication networks (ICNs) that enable multi-way communications among epithelial, fibroblast, endothelial, and immune cells. They also present that the capability of estimating individual ICNs reveals heterogeneity of ICNs and importance of intercellular communication in disease progression. The authors claim that the analyses of individual ICNs revealed structural patterns that were shared across subsets of tumors led to discovery of four different subtypes of tumor microenvironment (TME) networks of head and neck squamous cell carcinoma (HNSCC). The authors cleverly used the nested hierarchical Dirichlet processes model to the scRNAseq data to identify gene expression modules (GEMs) and employed gene set variation analysis to determine the expression status of the GEM in each tumor.

One major concern I have is how the authors claim that the patients with distinct TMEs exhibited significantly different clinical outcomes however the realistic clinical outcomes are not discussed. Below, I have listed a few questions and comments for the authors to address.

Major comments:

Although it is understood that the authors aimed to study the underlying TME associated events that are present in HNSCC (and as such can be detected with single cell analyses), the study design did not incorporate any commonly cogitated in clinic subtypes related to survival/response, i.e., subtypes classified by HPV status (cite: PMID: 34072836 or other bibliography).

1) Given that current challenges in the treatment of HNSCC include especially the HPV- patients, it would be advantageous to find out how the classes identified here (especially in relationship to the causal networks and immune response components) correspond to the HPV+ (with majority of patients responding well to treatment) vs. HPV- (patients will not do well).

2) By mining the patterns of ICNs of individual tumors, authors discovered distinct patterns of ICNs under TMEs of HNSCCs and claimed they were associated with significantly different patient outcomes. However, authors do not specify clinical details and known subtypes. Since the authors identified 39 immune GEMs and their signature genes, such analysis could benefit patients who would potentially respond to immune therapy and the value to this manuscript/work.

Minor comments:

Avoid jargon such as e.g., ‘paper’.

Reviewer #2: See attachment

Reviewer #3: In this work, Lu et al. develop a framework to individualized causal networks (ICNs) that describe intercellular communication relationships from single-cell RNA-sequencing data (scRNA-seq) sampled from patient tumors. The authors collected scRNA-seq data from 18 head and neck squamous cell carcinoma (HNSCC) patients. After identifying present cell types, the authors infer cell-type-specific gene expression modules (GEMs), sets of co-expressed genes that characterize cell state. From these GEMs, the authors use a variant of the famous Fast Causal Inference method, dubbed individualized Greedy Fast Causal Inference (iGFCI) to infer the ICNs, using Gene Structural Variant Analysis (GSVA) scores derived by deconvolving bulk RNA-seq data as proxy variables for GEMs. Both a global ICN for HNSCC and individualized ICNS are inferred, where the latter are inferred from 522 HNSCC tumors from the TGCA. Causal edges and trends in ICNS across patient groups are then analyzed.

While I think the ideas presented in this work are interesting and this work is a welcome step towards applying causal discovery methods to single-cell transcriptomics data, I think the results are difficult to interpret and several claims need to be validated in more detail. Therefore, I can only recommend publication after a major revision.

In particular, I would like the following comments to be addressed:

- The work would benefit from a more thorough comparison between the insights derived from analysis of patient-specific ICNs vs. the global ICN obtained for HSNCC. That is, what is gained or lost, in terms of biological insight, from analyzing only individual ICNs or analyzing only the global, “aggregated” ICN? Or, for example, could the individualized networks be used as additional prior knowledge when inferring the global network, or vice versa?

- Given that the inferred GEMs appear to vary with cell type, what is the difference between a GEM and, say, the set of marker genes inferred by differential expression analysis? From looking at the supplementary information, is it not immediately clear how a GEM differs from a set of differentially expressed genes.

- Did the authors verify whether or not all GEMs used in ICN inference actually contain cell signaling ligands? Otherwise, it may not be possible to claim that GEM_A -> GEM_B implies cell-cell communication. For example, at first glance, the top 20 genes of Imm GEM 1–5 do not contain signal ligands.

- If there are signal ligands, why not try use cell-cell communication inference tools (e.g., CellChat) for an analysis? At minimal, the authors should discuss the possibility of using such tools and their potential outcomes.

- The authors make the good point that current methods to infer cell-cell communication from scRNA-seq can only infer interactions that are known with respect to literature knowledge. Did the ICNs yield any novel insight, or did they confirm what was already known?

- It appears that the scRNA-seq is used to infer cell-type-specific GEMs, which is then used to deconvolve bulk RNA-seq data. The ICNs are then inferred using the deconvolved bulk RNA-seq data. Did the authors infer ICNs directly from the scRNA-seq data as well? Are there substantial differences between ICNs obtained from scRNA-seq data vs. ICNs obtained from bulk data?

- One claim is that testing if GEM_A -> GEM_B is equivalent to testing the relationship between two proxy variables, e.g. GSVA_A -> GSVA_B. The implication then is that these proxy variables are in one-to-one relationship with the GEMs. Did the authors check this?

- There’s an overall lack of interpretability behind the results presented in Figure 1¬–4. That is, while networks between GEMs are presented, there is a lack of discussion of what key genes are contained in these GEMs and what biological conclusions can be drawn. It would strengthen results to move some of the information from the Supplementary Figures.

- In general, the Figures could be improved by using text labels with larger font sizes, plot titles, and using vector graphics instead of PNG/JPEG files.

- In Figure 4, including color legends would really help a lot with interpretability of networks.

**Have the authors made all data and (if applicable) computational code underlying the findings in their manuscript fully available?**

Reviewer #1: **No: **Authors specify the data source. They don't specify the code(s) availability.

Reviewer #2: **No: **I do not see reference to the scRNAseq data analyzed (GSE or SRA #)

Reviewer #3: None

PLOS authors have the option to publish the peer review history of their article (what does this mean?). If published, this will include your full peer review and any attached files.

Reviewer #1: No

Reviewer #2: No

Reviewer #3: No
---

## [Decision Letter · Decision Letter 1]

7 Nov 2022

Dear Dr Lu,

Thank you very much for submitting your manuscript "An individualized causal framework for learning intercellular communication networks that define microenvironments of individual tumors" for consideration at PLOS Computational Biology. As with all papers reviewed by the journal, your manuscript was reviewed by members of the editorial board and by several independent reviewers. The reviewers appreciated the attention to an important topic. Based on the reviews, we are likely to accept this manuscript for publication, providing that you modify the manuscript according to the review recommendations.

The reviewers have overall found your changes and improvements satisfactory. Please address the remaining minor points by two of the reviewers.

Sincerely,

Philipp M Altrock, Ph.D.

Guest Editor

PLOS Computational Biology

Douglas Lauffenburger

Section Editor

PLOS Computational Biology

The reviewers have overall found your changes and improvements satisfactory. Please address the remaining minor points by two of the reviewers.

Reviewer's Responses to Questions

**Comments to the Authors:**

Reviewer #1: Thanks to the Authors for working on the revisions; the article is solid and ready for publication.

Reviewer #2: The authors addressed most of the criticisms. However, some minor concerns remain.

Some of the authors’ responses refer to changes and additions made in the revised version, but those were difficult to identify, and sometime were not to be found at all.

The major ones are listed below:

• In response to point (3), the authors state: “The results were from a 10-fold train-test experiments. We clarified this point in the Results (pp 14) and Methods (pp 21) sections.” I found the reference to 10-fold CV on pp. 14 and 21, but didn’t notice any acknowledgement of the the procedure’s caveat. The main criticism was that using the same model (i.e., the network structure learned on all data) is the main source of “information leakage” here when predicting a GEM from others (akin to performing feature selection outside the CV loop, which can significantly inflate accuracy), and this point is not addressed.

• In response to point (6), the authors state: “Thanks for pointing out the references. We have cited the references (pp 18) and discussed the differences from the methods.” However, if I look at p. 18 (or elsewhere), I couldn’t find the “cited references” nor a “discuss[ion of the] differences”.

The comparison to Celda should not be for this reviewer only. Rather, if this is a method paper, comparison of NHDP to other methods should be explicit part of the evaluation (given the manuscript is already pretty long, perhaps including this comparison in the supplementary document would be ok). The expanded description of NHDP does indeed help (although it remains scant for a method paper), but if the method is presented as a way of capturing “hierarchical” (lineage-like) structures, then comparison to (or at least discussion of) other methods aiming to do the same (e.g., K2Taxonomer, and trajectory-based methods such as Monocle, PAGA, etc.), and conjectures about why the new method should be preferable, should be included.

Reviewer #3: I am satisfied with the new manuscript and think the authors have done a good job in responding to all reviewer comments and making the paper clearer in its focus and potential biological applications. I therefore recommend that the paper be published now, with some additional minor comments that can be fixed easily.

Some minor comments:

- Please make sure the formatting of spacing, particularly between references and citations is correct, e.g. on page 4, ‘communications[1-6]’, and page 6, ‘(NHDP)[33]’, but also ‘top30’ on page 38

- Please check consistency in punctuation, e.g. ‘Almet et al. [25]’ vs ‘Armingol et al [26]’ on page 4 and ‘Obradovic et al [39]’ on page 12

- Figure 4c caption; as the subgraph contains only direct causal edges, I think you can remove the part about how to interpret blunt tailed edges.

**Have the authors made all data and (if applicable) computational code underlying the findings in their manuscript fully available?**

Reviewer #1: Yes

Reviewer #2: Yes

Reviewer #3: None

PLOS authors have the option to publish the peer review history of their article (what does this mean?). If published, this will include your full peer review and any attached files.

Reviewer #1: No

Reviewer #2: No

Reviewer #3: **Yes: **Qing Nie

Figure Files:

Data Requirements:

Reproducibility:

References:

---

## [Editor Report · Decision Letter 2]

26 Nov 2022

Dear Dr Lu,

We are pleased to inform you that your manuscript 'An individualized causal framework for learning intercellular communication networks that define microenvironments of individual tumors' has been provisionally accepted for publication in PLOS Computational Biology.

Sincerely,

Philipp M Altrock, Ph.D.

Guest Editor

PLOS Computational Biology

Douglas Lauffenburger

Section Editor

PLOS Computational Biology

---

## [Editor Report · Acceptance letter]

15 Dec 2022

PCOMPBIOL-D-22-00679R2 

An individualized causal framework for learning intercellular communication networks that define microenvironments of individual tumors

Dear Dr Lu,

I am pleased to inform you that your manuscript has been formally accepted for publication in PLOS Computational Biology. Your manuscript is now with our production department and you will be notified of the publication date in due course.

With kind regards,

Zsuzsanna Gémesi
